# Analysis of Infectious Complications after Thermal Ablation of Hepatocellular Carcinoma and the Impact on Long-Term Survival

**DOI:** 10.3390/cancers14215198

**Published:** 2022-10-23

**Authors:** Yutong Zhang, Xiaoju Li, Xiaoer Zhang, Tongyi Huang, Huanling Guo, Xiaoyan Xie, Chunyang Zhang, Ming Xu

**Affiliations:** Department of Ultrasound, The First Affiliated Hospital of Sun Yat-sen University, Guangzhou 510080, China

**Keywords:** hepatocellular carcinoma, thermal ablation, infection, recurrence, overall survival

## Abstract

**Simple Summary:**

Although thermal ablation has been widely used in treatments of HCC due to its unique advantages of less trauma, safety, postoperative complications still occur in clinical practice. Infectious complications are the third most common complication. The objective of our retrospective study was to summarize the clinical characteristics of infection after thermal ablation for HCC patients and to explore whether it affects tumor recurrence and overall survival. We enrolled 49 patients who developed postoperative infections and matched the same number of control patients, and we also summarized the clinical features and treatment of mild and severe infections. We found that postoperative infection among patients receiving ultrasound-guided thermal ablation adversely affected tumor progression. Empirical antibiotics and catheterization to reduce pressure inside the lesion should be utilized to minimize symptoms in patients with postoperative infection.

**Abstract:**

Purpose: This study aims to complete a detailed record of the clinical characteristics and treatment of HCC patients with post-ablation infection and evaluate the infections on recurrence-free survival (RFS) and overall survival (OS) among patients receiving ultrasound-guided thermal ablation. Methods: 3117 patients with liver tumors receiving thermal ablation from January 2010 to December 2021 were analyzed. A total of 49 patients with infectious complications after thermal ablation were selected as the infection group. A total of 49 patients without postoperative infection were randomly selected among those who underwent ablation within three days before or after the treatment date of the infection group as the control group. The clinical characteristics of both groups were analyzed by an independent sample t-test and chi-square test. A log-rank test was performed to compare the RFS and OS data. A multivariate Cox regression model was employed to identify prognostic factors influencing RFS and OS. Subgroup analyses of mild and severe infections were conducted to explore the infection-related situation further. Results: Between mild and severe infection groups, there were statistically significant differences in the infection position (*p* = 0.043), positive rate of body fluid culture (*p* = 0.002), proportion of catheter drainage (*p* = 0.017), use of advanced antibiotics (*p* = 0.006), and outcome (*p* = 0.00). The Kaplan–Meier survival analysis revealed that postoperative infection was significantly correlated with tumor recurrence (*p* = 0.028), and severe infection was significantly associated with overall survival (*p* = 0.049). The cox model showed that postoperative infection was an independent variable for RFS deterioration (HR = 1.724, 95% CI: 1.038–2.862, *p* = 0.035). Conclusions: Postoperative infection among patients receiving ultrasound-guided thermal ablation adversely affected tumor progression. In addition, empirical antibiotics and catheterization to reduce pressure inside the lesion should be utilized to minimize symptoms in patients with postoperative infection.

## 1. Introduction

Hepatocellular carcinoma (HCC) is the fourth leading cause of cancer-related death in the world [1]. The treatment of HCC mainly includes partial hepatectomy, liver transplantation, thermal ablation, alcohol ablation, transhepatic arterial chemoembolization, chemotherapy, immunotherapy, and combination therapy [2,3]. Thermal ablation has been widely used in the treatment of early HCC and secondary liver malignancies due to its unique advantages of less trauma, safety, low mortality, short hospitalization, and low cost [4,5,6,7]. Although thermal ablation is regarded as a safe procedure, postoperative complications still occur in clinical practice [8,9,10]. Infectious complications are the third most common complication after tumor spread and abdominal bleeding during the ablation treatment [11].

There are many types of post-ablation infection, including liver abscess, biliary tract infection, peritonitis, soft tissue abscess, respiratory tract infection, urinary tract infection, sepsis, and unexplained causes. Infection not only increases the physical and economic burden of patients but also has a negative impact on tumor prognosis and overall survival. Studies show that inflammation can increase the adhesion of circulating tumor cells or bacterial antigens that can directly increase the ability of cancer cells to metastasize, thus accelerating tumor metastasis [12,13]. However, few studies report clinical characteristics after post-ablation infection in detail, and no study has further focused on whether the infection has an impact on patient prognosis.

This study aims to complete a detailed record of the clinical characteristics and treatment of HCC patients with post-ablation infection in our center in the past 12 years, as well as to assess the impact of postoperative infectious complications on recurrence-free survival (RFS) and overall survival (OS) in patients receiving ultrasound-guided thermal ablation for HCC.

## 2. Patients and Methods

The study was approved by the Ethics Committee of the First Affiliated Hospital of Sun Yat-Sen University (Guangzhou, China). Since it was a retrospective study, informed consent of the patients could not be obtained, but all the patients had signed informed consent before the ablation.

### 2.1. Patient Selection

From January 2010 to December 2021, 3117 patients with liver tumors were treated with ultrasound-guided thermal ablation at the First Affiliated Hospital of Sun Yat-sen University (Guangzhou, China), and post-ablation infection occurred in 83 patients. Thermal ablation includes radiofrequency ablation and microwave ablation. The definition of infectious complications after percutaneous thermal ablation is defined as follows: The infection occurred within 30 days after the ablative procedure. In addition, it must meet at least one of the following: (a) Microbiological demonstration: Organisms isolated from blood specimens or an aseptically obtained culture of fluid or tissue in the organ/space. (b) Visible purulent drainage or secretion. (c) An abscess or other evidence of infection in-volving the organ/space that is found on a radiologic examination. (d) Clinical diagnosis: Clinical symptoms and signs (persistent moderate to high fever, i.e., body temperature > 38.5 °C and lasting for more than 3 days, with or without chilling), laboratory examination (WBC > 10 × 10^9^/L and showed an upward trend or PCT > 0.5 ng/mL); antibiotics were effective. Inclusion criteria were as follows: 1. Primary hepatocellular carcinoma was the target of ablation. 2. The infection occurred within 30 days after the ablative procedure. The exclusion criteria were as follows: 1. Combined with history, imaging, or pathology, the liver malignancy was not a primary hepatocellular carcinoma. 2. Follow-up was less than 6 months. 3. Lymph node metastasis had already occurred preoperatively.

A total of 49 patients were recruited to the infection group. The control group was randomly selected among an equal number of patients who underwent thermal ablation of HCC within three days before and after the treatment date of the infection group but did not develop an infection. All patients, including the infection group and the control group, were treated with single thermal ablation without fractional ablation (Figure 1).

### 2.2. Thermal Ablation

The thermal ablation was decided by the multidisciplinary team (MDT) including surgeons and radiologists. All ablations were performed by interventional doctors with at least 5 years of experience in liver tumor ablation. Before ablation, an ultrasound contrast examination was performed to determine the tumor location and safe boundary, and then vertical needle insertion was performed. Artificial ascites were used for protection when the tumor was near the gastrointestinal tract, diaphragm, and other easy-to-damage other parts. 

Ablation devices and experience evolved in our center over the study period from 2010 to 2021. As for radiofrequency ablation (RFA) systems, we used Starburst Talon (RITA Medical Systems, Mountain View, CA, USA), the RF 2000 system (RadioTherapeutics, Sunnyvale, CA, USA) with an expandable electrode, and the internally cooled electrode system (Valleylab, Covidien/Medtronic, Minneapolis, MN, USA) with a unipolar Cool-tip electrode. The microwave ablation (MVA) systems with cooling antennas were manufactured by two companies: The KY-2000 MWA system (Canyon Medical, Nanjing, China) and the Emprint ablation system (Covidien/Medtronic, Minneapolis, MN, USA). The choice of ablation devices was made by the individual interventional radiologists.

The ablation was performed with conscious sedation (0.1 mg of fentanyl, 5 mg of droperidol, 0.1 mg of tramadol hydrochloride, IV) and local anesthesia (5 mL of 1% lidocaine). Vital signs were continuously monitored during the procedure. Standard sterile techniques were applied, including hand washing, skin disinfection, surgical gown, gloves, and drapes. 

B-mode images and contrast-enhanced ultrasound images during ablation were recorded.

### 2.3. Information Collection

We collected the patients with baseline data (age, gender, past medical history, whether cirrhosis liver disease, Child-Pugh class, laboratory results, etc.), infections (peak temperature, the infection position, presence of hemodynamic changes, results of body fluids culture, etc.), treatment (application and duration of antibiotics, location, and duration of catheter drainage, ever into ICU, hospitalization days, patient’s outcome, etc.), and follow-up information (time of the first postoperative recurrence, recurrence type, overall survival time and cause of death). All patients underwent a single and thorough intervention episode including one or more ablations of one or more tumors, and the efficacy of ablations was assessed by contrast-enhanced ultrasound on and around the ablation site the next day. The above data were divided into the severe infection group and mild infection group according to the occurrence of hemodynamic changes.

### 2.4. Follow-Up and Data Analysis

Recurrence is divided into local tumor progression, intrahepatic recurrence, and distant metastasis. Local tumor progression (LTP) was defined as the appearance of viable cancer tissue touching the initially treated tumor [14]. Intrahepatic recurrence refers to new intrahepatic lesions that are not in contact with the initially treated tumor. Distant recurrence is the emergence of one or several tumor(s) separate from the primary site.

After HCC thermal ablation, regular follow-up should be conducted at 1, 3, 6, 9 months with contrast-enhanced ultrasound (CEUS) and at least 2 computerized tomography (CT) and magnetic resonance imaging (MRI) procedures within 1 year. Recurrence-free survival was measured from the time of thermal ablation to the period when the first recurrence was detected. The data of patients diagnosed with recurrence for the first time after ablation can be obtained in the following two ways: 1. Patients regularly reviewed in our hospital with clear imaging evidence, such as CT/MRI or CEUS. 2. For patients who did not have a definite recurrence in the last examination of our hospital, we would contact the patients or their family members by phone to confirm the time of the first recurrence. The patient’s overall survival was defined as the time from the date of thermal ablation to the date of death (due to recurrence or any other cause). The survival time of the patients was also verified by the patients themselves or their family members. All time estimates were made from the date of ablation at which infection occurred. The follow-up was finalized at either death or the last visit to the outpatient clinic before 1 November 2021. 

### 2.5. Statistical Analysis 

The independent sample t-test and chi-squared test were used to compare continuous variables and categorical variables between the infection and control groups. The data are presented as the mean ± standard deviation or median ± range if the data were not normally distributed. The Kaplan–Meier method was used for univariate analysis, and the results were compared using the log-rank test. The subgroup analysis of the prognosis of mild and severe infections was the same as before. Using the statistically significant variables obtained through univariate analysis, a Cox proportional-hazards model was then applied to perform. Variables were significant when *p* values were <0.05. The statistical analyses were performed using the IBM SPSS 25.0 Statistics software (Armonk, NY, USA: IBM Corp), and the survival curve was generated by R (version 3.6.1). 

## 3. Result

### 3.1. Basic Characteristics

Among 3117 cases of liver cancer thermal ablation, 49 patients (43 men and 6 women, mean age, 59.1 ± 12.8 years) were selected for the postoperative infection group, including 8 patients with severe infection and 41 patients with mild infection. The control group was matched with 49 patients, including 47 men and 2 women (mean age, 57.4 ± 12.1 years). The characteristics of the included patients are listed in Table 1. 

There were statistical differences in AST, tumor number, tumor maximum diameter, and TACE history between the infection group and the control group, and no significant statistical differences in other data between the two groups.

Most of the patients in both groups had cirrhosis, and about a quarter had diabetes before surgery. The preoperative ALT level was 40.4 ± 38.4 U/L, and ASL was 50.1 ± 54.5 U/L for patients in the infection group, which were higher than ALT (28.7 ± 15.0 U/L) and AST (32.5 ± 15.7 U/L) in the control group. The number of tumors was 1.9 ± 0.9 in the infection group, which was higher than that of the control group (1.4 ± 0.7). The maximum tumor diameter was also larger in the infected group (2.7 ± 1.3 cm) than in the control group (2.2 ± 0.8 cm). A total of 19 patients (38.8%) previously underwent TACE surgery, compared with only 7 patients (14.3%) in the control group. The preoperative Child-Pugh class in both groups was mainly A; AFP significantly increased; and most patients were treated with radiofrequency ablation.

### 3.2. Clinical Manifestation with Postoperative Infection

The characteristics of the mild and severe infection patients are listed in Appendix A. There was no statistical difference between the two groups at baseline. Details of mild and severe infections and their treatment are presented in Table 2. 

There were significant differences in the infection position (*p* = 0.043), positive rate of body fluid culture (*p* = 0.002), proportion of catheter drainage (*p* = 0.017), use of advanced antibiotics (*p* = 0.006), and outcome (*p* = 0.00) between mild and severe infection groups. Patients with mild infection developed an infection 3.9 ± 6.7 days after ablation, and the mean thermal spike was 39.3 ± 0.5 °C. Intrahepatic infection accounted for a large proportion of mild infection patients (87.8%), and the positive rate of body fluid culture results was 24.4%. Among 41 patients, 5 were treated with advanced antibiotics, such as imipenem and vancomycin, and the rest were treated with ordinary antibiotics, such as cephalosporins and quinolones; 10 patients were treated with catheter drainage. All patients with mild infection were cured and discharged, with an average hospital stay of 17.3 ± 20.1 days. 

Patients with severe infection developed an infection 7.6 ± 10.4 days after ablation, and the thermal spike of patients (38.8 ± 1.0 °C) was less than that of patients with mild infection, but the differences between patients were greater. Among patients with severe infection, there was no difference between intrahepatic infection (50.0%) and extrahepatic infection (50.0%), among which there was 1 case of liver abscess, 3 cases of biliary tract infection, and 4 cases of peritoneal infection. The positive rate of body fluid culture was higher (87.5%). A total of 5 of the 8 patients received advanced antibiotics, and 6 patients were treated with catheter drainage. A total of 5 patients were cured and discharged from the hospital. A total of 3 patients died, and the average length of stay was 21.3 ± 5.8 days. 

Body fluid samples were mostly obtained from blood, drainage fluid, urine, and sputum. In mild cases, 30 patients were negative in the body fluid culture, accounting for 73.1%, and, in severe cases, only 1 patient was negative, accounting for 12.5%. The more common positive results were Escherichia coli, Klebsiella pneumonia, Staphylococcus, fungi, and other rare bacteria. The culture results in detail are shown in Figure 2.

Catheter drainage was performed in 16 patients with postoperative infection, including 10 mild cases (62.5%) and 6 severe cases (37.5%). Most of these were placed in intrahepatic abscesses, the intrahepatic bile duct, and the subcutaneous abdominal wall. The days of catheter drainage were evaluated according to the patient’s symptoms, drainage situation, and ultrasound images. The changes in body temperature and WBC of 14 patients before and after catheterization were collected (data of 2 patients were lost). The values of body temperature and WBC before catheterization were 39.0 ± 0.6 °C and 14.2 ± 6.6 × 10^9^/L, respectively. After catheterization, the temperature and WBC values were 36.7 ± 0.2 °C and 6.3 ± 2.6 × 10^9^/L, respectively. There were significant differences in body temperature (*p* = 0.00) and leukocytes (*p* = 0.01).

### 3.3. Recurrence and Survival

At the time of the follow-up in the infection group, 33 (67.3%) patients developed tumor recurrence; 11 (22.4%) patients died of HCC progression; and 7 (14.3%) patients died of non-cancerous disease. In the control group, 29 (59.2%) patients developed tumor recurrence; 15 (30.6%) patients died of HCC progression; and 2 (4.1%) patients died of non-cancerous disease. The 5-year RFS rate in the infection group and the control group was 30.7% and 40.8%, and the 5-year OS rate was 63.3% and 65.3%, respectively. The details of recurrence and overall survival of postoperative infected patients are shown in Table 3. The types of recurrences in the infection group included 3 LTPs, 25 intrahepatic recurrences, 2 new distant metastases, 1 with local and intrahepatic recurrences meanwhile, and 2 with intrahepatic recurrence and new distant metastases meanwhile. The types of recurrences in the control group included 6 LTPs, 25 intrahepatic recurrences, and 2 new distant metastases.

The Kaplan–Meier survival analysis showed that postoperative infection was significantly correlated with tumor recurrence (*p* = 0.028), although no significant difference was detected between mild and severe groups. The results are shown in Figure 3.

The results of the univariate and multivariate COX regression analysis that analyzed the relationship between multiple clinical characteristics and recurrence are shown in Table 4. Postoperative infection is an independent factor for recurrence (*p* = 0.031). Although hepatic cirrhosis and diabetes did not reach statistical significance on the univariate analysis, they were also included in the multivariate analysis. On the multivariate analysis, postoperative infection (*p* = 0.035) was still the influencing factor for tumor recurrence.

There was no statistically significant difference between the infection and control group on overall survival (*p* = 0.46) on the Kaplan–Meier survival analysis, but, in the subgroup analysis, we found that patients with severe infection had significantly lower overall survival than those without infection (*p* = 0.049), while mild infection had no significant effect on overall survival. The results are shown in Figure 4. Further univariate and multivariate COX regression analyses showed that infection after thermal ablation was not an independent predictor of overall survival (*p* = 0.369; *p* = 0.318). The results of clinical characteristics and OS are shown in Appendix A.

## 4. Discussion

This study summarized the clinical characteristics of patients with HCC after thermal ablation infection in our center; data from a total of 49 patients found that intrahepatic infection is the most common type of infection including liver abscess, ablative focal gas, biliary tract infection, biliary duct injury, etc. Patients with severe infections are more likely to have a culture positive for pathogens in their body fluids, while *E. coli* and klebsiella pneumonia bacteria are common pathogens for both mild and severe infections. In the selection of infection treatment, most patients with the severe infection will be combined with catheter drainage and advanced antibiotics to relieve the symptoms of infection. In addition to increasing the length of hospital stay, patients with mild infection can be discharged after recovery, while the occurrence of severe infection will have a relatively large negative impact on the survival of patients. Postoperative infection of HCC is more likely to cause tumor recurrence, and severe infection will affect the overall survival of patients.

In the retrospective study of 3117 patients with liver tumors who underwent thermal ablation including radiofrequency and microwave ablation, 83 patients with postoperative infection were found in this study, with an incidence of 2.7%. According to a systematic review, liver abscess occurs in 0.32% of patients who undergo ablative techniques for liver tumors [15]. In this study, the type of infection after liver tumor thermal ablation is not confined to the liver abscess and also includes biliary infection, peritoneal cavity infection, infection of the respiratory system, urinary system infection, central catheter infections, skin infections, and no clear focal infection was found, but there was a clear existence of infection in patients with clinical evidence, so the overall infection rate in this study is higher. In general, thermal ablation is a relatively safe and effective operation for patients with early HCC or patients with poor basic conditions who cannot tolerate surgery. The incidence of postoperative complications ranged from 0 to 6.1%, much lower than surgical excision [16,17,18,19].

Intrahepatic infection including liver and biliary tract infection is one of the common complications of thermal ablation; it may be related to the cause of the patient’s medical history (diabetes, cirrhosis of the liver caused by abnormal immune globulin, history of biliary tract surgery, surgery with conventional history), tumor conditions (diameter size, location), or surgical aseptic conditions [20,21]. In this study, there were 13 diabetic patients, 6 of whom had preoperative blood glucose > 6 mmol/L; 9 patients had a history of biliary tract surgery, including choledochostomy, choledochotomy, cystectomy, percutaneous liver puncture biliary drainage, and other surgical procedures focusing on large bile ducts, which may cause bile duct injury and stenosis. The larger the total volume of the tumor, the longer the ablation operation time and the larger the surgical scope. If the tumor is near the intestine, biliary tract, and diaphragm, it is more likely to damage the surrounding organs, and the probability of infection will be greater. In previous studies, TACE operation history and tumor diameter were risk factors for postoperative infection. Another related study by our group also confirmed this point. TACE blocked intrahepatic blood flow, had a large ablation area, and demonstrated slow heat dissipation. The tumor diameter is large; the tumor center is necrotic; and ablation is more likely to damage the large bile duct [22].

In addition, the intestinal mucosal barrier function was reduced, and permeability was increased in patients with cirrhosis. In our study, the body fluid culture positive for *E. coli* and klebsiella pneumonia bacteria is relatively usual; Common Escherichia coli and Klebsiella pneumoniae in the intestinal lumen [23,24] are prone to ectopic colonization through the lymph or portal vein, resulting in an increased risk of infection. Because Gram-negative bacteria are the most common bacteria infected after ablation, in the early stage of postoperative infection, antibiotics against such bacteria can be empirically used, such as the second- and third-generation cephalosporin antibiotics, quinolone antibiotics, penicillin, and most infections can be controlled. If infection control is poor and clinical infection indicators or patients’ conditions are still not alleviated, upgraded antibiotics can be used, including imipenem, meropenem, vancomycin, etc., and corresponding antibiotics can be applied after the results of bacterial culture are available. In the process of infection treatment, imaging methods should be used to find the source of infection. If focal infection foci can be clearly found, such as liver abscess, biliary tract infection, and thorax and abdominal effusion, drainage tubes can be placed to reduce the absorption of bacterial toxins and accelerate the relief of infection and poisoning symptoms. In the 16 patients with catheterization, the temperature and white blood cells decreased significantly after catheterization, which encouraged doctors to catheterization and drainage in the treatment of patients with infection after ablation. In this study, the proportion of patients with severe infections who underwent catheterization was significantly higher than that of patients with mild infection, and patients with catheterization had longer hospital stays, which may be because the infection of patients with catheterization was more serious.

For how to prevent infection at the surgical site, the Centers for Disease Control and Prevention recommends prophylactic antibiotics one hour before surgery followed by additional injections every three to four hours throughout the procedure; extension of antibiotic use beyond 24 h is not recommended [25,26]. A randomized controlled trial showed that a single use of prophylactic antibiotics within 24 h before ablation can reduce the incidence of infectious complications [27]. In this study, 13 people in the mild infection group received prophylactic antibiotics before surgery, mainly cephalosporins, and 2 people in the severe infection group. Although there are few controversial studies on infection prevention, however, preoperative improvement in liver function, blood glucose control, short-term targeted use of antibiotics against Gram-negative bacteria, and strict control of aseptic surgical conditions are beneficial for patients in reducing the risk of postoperative infection. We believe that the occurrence of infection after ablation is likely to be related to the surgical environment, the experience of the surgeon, and the batch of equipment, so random matching within three days before and after the ablation date was selected in this study instead of using Propensity score matching (PSM).

Infection after hepatectomy has been reported to affect tumor-related prognosis negatively, and postoperative infection is an independent predictor of disease-free survival and overall survival [28,29]. At present, there is almost no systematic report and summary of the impact of infection after ultrasound-guided thermal ablation on the prognosis of patients. In this study, patients with infection after thermal ablation were followed up, and it was found that infection also had a negative impact on tumor recurrence, and severe infection significantly shortened the overall survival time of patients. In our study, both KM analysis and COX analysis showed that infection after thermal ablation was an independent influencing factor for RFS, but the degree of postoperative infection did not seem to have a significant impact on tumor recurrence. Infection and the stress response can lead to an excessive synthesis and release of proinflammatory cytokines, such as interleukin IL-1, IL 6, and TNF alpha; studies show that these cytokines can inhibit other immune active circulating cells, such as cytotoxic T lymphocytes, natural killer cells, and dendritic antigen-presenting cells to induce immunosuppression [28]. The relationship between postoperative infection and tumor recurrence is complex and still under investigation. In the vast majority of patients with primary liver cancer merger cirrhosis, an unbearable degree of postoperative infection area is large, and heavy infection, hemodynamic changes tend to be more prone to a poor prognosis. Therefore, how to reduce the incidence of infection in patients after thermal ablation deserves the attention of clinicians; especially for patients with high-risk factors of infection, preoperative prophylactic use of antibiotics, strict adherence to the principle of sterility during surgery, and other measures are particularly important. In a meta analysis, the 5-year recurrence rate of radiofrequency ablation was 77.7%, and the 5-year survival rate was 50.5% [30]. The 5-year survival rate in this study was slightly higher than that in previous studies, which may be due to the number of patients lost to follow-up due to the long years, or the patients enrolled in a single center.

The limitations of this study are as follows: 1. This study is a retrospective study with selection bias and a long period. Indications for thermal ablation are constantly changing, and surgical equipment, experience, and techniques improve over time, making it difficult to control variables uniformly. 2. Data were obtained from a single center in a hospital in Asia, and differences in indications, methods, expertise, performance of available ultrasound and CT equipment, and nursing ability of HCC thermal ablation may make the study results impossible to reproduce in other zones or centers. 3. Thermal ablation is a relatively safe operation with fewer complications and a low infection rate. Although we collected patients with postoperative infection in our center in the past 12 years, the number of cases included is still small.

In conclusion, ultrasound-guided thermal ablation is a safe and effective treatment for HCC with a low incidence of postoperative infection. Postoperative infection adversely affected tumor progression. For patients with postoperative infection, empirical use of antibiotics and catheterization actively for the source of infection should be taken to relieve the symptoms of infection as much as possible.

## Figures and Tables

**Figure 1 cancers-14-05198-f001:**
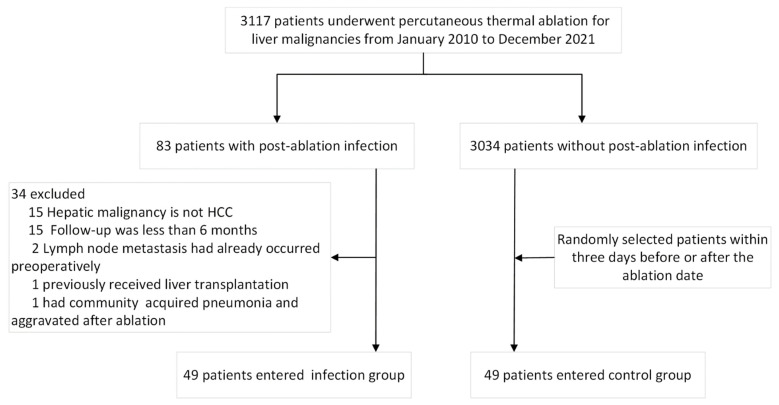
Flow chart showing the selection of study patients.

**Figure 2 cancers-14-05198-f002:**
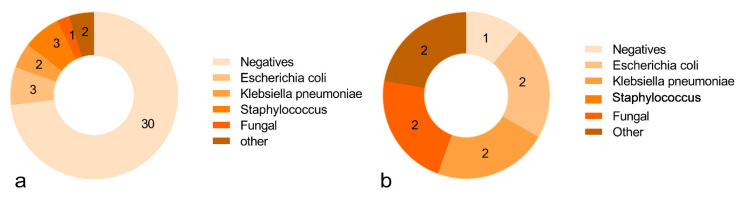
The culture results of patients with mild (**a**) and severe (**b**) infection. The numbers in the figure represent the number of people. There were 45 cases of mild infection and 9 cases of severe infection, which were inconsistent with the number of cases included in the study, because there were two patients with multiple infections in each of the severe and mild patients, respectively, infected with Escherichia coli and Klebsiella pneumoniae, so the culture results were counted separately in this study.

**Figure 3 cancers-14-05198-f003:**
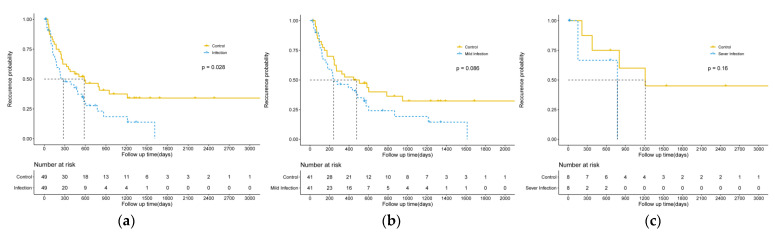
Graphs display the recurrence curves for HCC patients receiving ultrasound-guided thermal ablation with and without postoperative infection. ((**a**): All infected group and control group; (**b**): Mild infection group and control group; (**c**): Severe infection group and control group).

**Figure 4 cancers-14-05198-f004:**
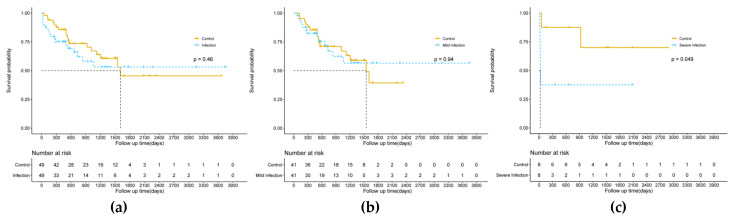
Graphs display the overall survival curves for HCC patients receiving ultrasound-guided thermal ablation with and without postoperative infection. ((**a**): All infected group and control group; (**b**): Mild infection group and control group; (**c**): Severe infection group and control group).

**Table 1 cancers-14-05198-t001:** Baseline data for the infected and control groups.

	Infection Group (49 Patients)	Control Group (49 Patients)	*p* Value
Age (years).	59.1 ± 12.8	57.4 ± 12.1	0.492
Gender (M/F)	87.8% (43/6)	95.9% (47/2)	0.268
Hepatic Cirrhosis (Y/N)	87.8% (43/6)	91.8% (45/4)	0.739
Diabetes (Y/N)	26.5% (13/36)	22.4% (11/38)	0.638
HBsAg (+) (Y/N)	87.8% (43/6)	87.8% (43/6)	1.000
Child-Pugh class			1.000
A	93.9% (46)	93.9% (46)	
B	6.1% (3)	6.1% (3)	
Alanine aminotransferase level (U/L)	40.4 ± 38.4	28.7 ± 15.0	0.051
Aspartate aminotransferas Level (U/L)	50.1 ± 54.5	32.5 ± 15.7	0.034 *
Total bilirubin Level (mg/dL)	18.4 ± 9.7	18.0 ± 12.0	0.851
Alpha-fetoprotein Level (ug/L)	1523.7 ± 6361.9	1302.0 ± 3529.8	0.835
Thermal ablation type			1.000
Radiofrequency ablation	91.8% (45)	89.8% (44)	
Microwave ablation	8.2% (4)	10.2% (5)	
Tumor number	1.9 ± 0.9	1.4 ± 0.7	0.004 *
Tumor size (maximum diameter)	2.7 ± 1.3	2.2 ± 0.8	0.022 *
Transcatheter arterial chemoembolization (Y/N)	38.8% (19/30)	14.3% (7/42)	0.006 *

Y: yes; N: no, * Significant difference. Data are expressed as mean ± s.d.

**Table 2 cancers-14-05198-t002:** Clinical characteristics of patients with mild and severe infection.

	Mild Infection	Severe Infection	*p* Value
Number	41	8	
Age (years)	58.2 ± 13.0	63.8 ± 10.9	0.267
Gender (M/F)	35/6	8/0	0.571
The interval between ablation and infection (d)	3.9 ± 6.7	7.6 ± 10.4	0.195
Thermal spike (°C)	39.3 ± 0.5	38.8 ± 1.0	0.159
Infections position			0.043 *
intrahepatic	87.8% (36)	50.0% (4)	
extrahepatic	12.2% (5)	50.0% (4)	
Shiver (Y/N)	26.8% (11/30)	25% (2/6)	1.00
Bacterial culture (P/N)	24.4% (10/31)	87.5% (7/1)	0.002 *
Catheterization (Y/N)	24.4% (10/31)	75.0% (6/2)	0.017 *
Advanced antibiotics (Y/N)	12.2% (5/36)	62.5% (5/3)	0.006 *
Hospital stays (d)	17.3 ± 20.1	21.3 ± 5.8	0.589
Outcome			0.00 *
recovery	100% (41)	62.5% (5)	
death	0% (0)	37.5% (3)	

M: male; F: female; Y: yes; N: no; P: positive; N: negative; d: days; °C: degree centigrade; * Significant difference. Data are expressed as mean ± s.d.

**Table 3 cancers-14-05198-t003:** Recurrence and Survival of patients infected after thermal ablation.

		Recurrence-Free Survival (%)			Overall Survival (%)		
	*n*	1-Year	2-Year	5-Year	1-Year	2-Year	5-Year
infection group	49	55.1 (27)	40.8 (20)	30.7 (16)	73.5 (36)	63.7 (33)	63.3 (31)
mild infection	41	48.8 (20)	31.7 (13)	24.4 (10)	82.9 (34)	73.2 (30)	68.3 (28)
severe infection	8	87.5 (7)	87.5 (7)	75.0 (6)	37.5 (3)	37.5 (3)	37.5 (3)
control group	49	59.2 (29)	49.0 (24)	40.8 (20)	85.7 (42)	77.6 (38)	65.3 (32)

**Table 4 cancers-14-05198-t004:** Univariate and multivariate COX regression analysis for RFS.

Characteristics	Univariate Analysis		Multivariate Analysis	
	HR (95%CI)	*p* Value	HR (95%CI)	*p* Value
Age	1.172 (0.710–1.933)	0.535		
<60 year				
≥60 year				
Gender	0.656 (0.311–1.381)	0.267		
Male				
Female				
Hepatic Cirrhosis	0.847 (0.364–1.968)	0.699	0.887 (0.375–2.095)	0.784
No				
Yes				
HBsAg (+)	0.712 (0.350–1.446)	0.347		
No				
Yes				
Child-Pugh class	0.592 (0.185–1.890)	0.376		
A				
B				
Diabetes	1.363 (0.776–2.393)	0.281	1.287 (0.725–2.286)	0.390
No				
Yes				
Alpha-fetoprotein (ug/L)	1.617 (0.977–2.675)	0.061		
<20				
≥20				
Thermal ablation type	0.407 (0.128–1.300)	0.130		
Radiofrequency ablation				
Microwave ablation				
Postoperative infection	1.745 (1.054–2.891)	0.031	1.724 (1.038–2.862)	0.035
No				
Yes				

## Data Availability

Not applicable.

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
