# Peer review of "Analysis of Infectious Complications after Thermal Ablation of Hepatocellular Carcinoma and the Impact on Long-Term Survival"

_cancers, 2022, doi:10.3390/cancers14215198_

Round 1

Reviewer 1 Report

3 days of control group (what is control group here)?

Paragraph 1, last line: Infectious complication in HCC patients in general? Or in HCC patients with thermal ablation?

Page 2, last line: Define RFA at first use, before abbreviating

Did the complication group also have single ablation without fractional ablation? Does that affect the overall survival?

More description of recurrence-free survival and OS in text will be helpful

No description of results from table 4 was written in the text. More discussion on this will help the reader.

There were no clear individual references for Figure 3a, b, c, the whole Figure 3. Similar comments for Figure 4. There are instances of indicating p value in some places and did not indicate in other places in the text. Be consistent.

Reviewer 2 Report

Hepatocellular carcinoma (HCC) is one of the leading causes of cancer-related deaths worldwide. It currently represents the fifth most common malignancy among men and the seventh in women, with an increasing incidence due to which HCC is supposed to become the third leading cause of cancer-related deaths in Western Countries by 2035. Radical removal of the tumor is the only potentially curative treatment for HCC; however, more than half of HCC cases are detected when advanced (locally advanced / unresectable or metastatic) and the majority of patients can receive only palliative treatments, with poor prognosis and short life expectancy.

A number of staging and classification systems have been used and proposed for HCC, in order to define the “double-headed” nature of the disease, with prognosis and treatment depending on tumor burden, hepatic functional reserve and patient’s underlying liver disease. Among them, the Barcelona Clinic Liver Cancer Classification (BCLC) is probably the most widely used worldwide; according to this system, very early (BCLC-0) and early-stage (BCLC-A) HCC are amenable to curative treatments, including radical surgical resection, liver transplantation and tumor ablation. The possibility to obtain complete removal of the tumor at this stage has led to the development of several minimally invasive treatment options including radiofrequency ablation (RFA), microwave ablation (MWA), cryoablation and percutaneous ethanol injection (PEI). Moreover, the combination of systemic treatments with locoregional techniques has been also investigated and there are currently ongoing trials aimed to evaluate efficacy and safety of multimodal treatments. Local ablative strategies such as percutaneous RFA and percutaneous MWA have shown to be safe and feasible in HCC, where RFA represents the most frequently used ablative modality in unresectable BCLC-A HCC, reaching outcomes comparable to those of surgical resection in case of single nodules less than 2 cm in size.

Some changes are required, including a linguistic revision.

- We believe this article is suitable for publication in the journal although major revisions are needed. The main strengths of this paper are that it addresses an interesting and very timely question and provides a clear answer, with some limitations. Certainly, the study is limited to an Asian population with a relatively very small sample size, and the authors should further express this point.
- Second, the study included a widely varied patient population and the total number of patients analyzed was relatively small. Thus, the authors should better highlight the limitations of the current paper.
- The background of the changing scenario of medical treatment in HCC should be better discussed, and some recent papers regarding this topic should be included
.

Major changes are necessary.

Round 2

Reviewer 2 Report

acceptance